# The Convergence of Radiology and Genomics: Advancing Breast Cancer Diagnosis with Radiogenomics

**DOI:** 10.3390/cancers16051076

**Published:** 2024-03-06

**Authors:** Demetra Demetriou, Zarina Lockhat, Luke Brzozowski, Kamal S. Saini, Zodwa Dlamini, Rodney Hull

**Affiliations:** 1SAMRC Precision Oncology Research Unit (PORU), DSI/NRF SARChI Chair in Precision Oncology and Cancer Prevention (POCP), Pan African Cancer Research Institute (PACRI), University of Pretoria, Hatfield, Pretoria 0028, South Africa; dd.demetriou@up.ac.za; 2Department of Radiology, Faculty of Health Sciences, Steve Biko Academic Hospital, University of Pretoria, Hatfield, Pretoria 0028, South Africa; zarina.lockhat@up.ac.za; 3Translational Research and Core Facilities, University Health Network, Toronto, ON M5G 1L7, Canada; luke.brzozowski@uhn.ca; 4Fortrea Inc., 8 Moore Drive, Durham, NC 27709, USA; kamalveer.saini@fortrea.com; 5Addenbrooke’s Hospital, Cambridge University Hospitals NHS Foundation Trust, Cambridge CB2 0QQ, UK

**Keywords:** breast cancer, prognosis, molecular heterogeneity, radiogenomics, precision medicine, early diagnosis, treatment options, clinical behaviors, localized research, global disparities

## Abstract

**Simple Summary:**

Significant strides have been made in the understanding and management of breast cancer (BC), yet it remains a pervasive and life-threatening illness globally, necessitating precise diagnostic and therapeutic approaches. Molecular subtyping of BC is crucial for prognostic and predictive purposes due to its diverse clinical behaviors. Disparities in diagnosis and outcomes across different populations, particularly among individuals of African heritage, underscore the importance of localized research efforts. Early detection and treatment are imperative for improving survival rates, prompting the emergence of radiogenomics within precision medicine as a promising avenue. Radiogenomics, which integrates genetic patterns with imaging features, holds potential for enhancing early detection, prognosis, and treatment selection. By eliminating the need for biopsy and sequencing, it streamlines clinical workflows and contributes to advancing individualized patient care. However, challenges such as reproducibility, standardization, and data integration remain, highlighting the need for further research and validation, particularly in prospective multi-institutional settings. Despite limitations, radiogenomics stands as a promising frontier in breast cancer research, with the potential to revolutionize patient care through its non-invasive approach and ability to correlate genomic information with imaging phenotypes.

**Abstract:**

Despite significant progress in the prevention, screening, diagnosis, prognosis, and therapy of breast cancer (BC), it remains a highly prevalent and life-threatening disease affecting millions worldwide. Molecular subtyping of BC is crucial for predictive and prognostic purposes due to the diverse clinical behaviors observed across various types. The molecular heterogeneity of BC poses uncertainties in its impact on diagnosis, prognosis, and treatment. Numerous studies have highlighted genetic and environmental differences between patients from different geographic regions, emphasizing the need for localized research. International studies have revealed that patients with African heritage are often diagnosed at a more advanced stage and exhibit poorer responses to treatment and lower survival rates. Despite these global findings, there is a dearth of in-depth studies focusing on communities in the African region. Early diagnosis and timely treatment are paramount to improving survival rates. In this context, radiogenomics emerges as a promising field within precision medicine. By associating genetic patterns with image attributes or features, radiogenomics has the potential to significantly improve early detection, prognosis, and diagnosis. It can provide valuable insights into potential treatment options and predict the likelihood of survival, progression, and relapse. Radiogenomics allows for visual features and genetic marker linkage that promises to eliminate the need for biopsy and sequencing. The application of radiogenomics not only contributes to advancing precision oncology and individualized patient treatment but also streamlines clinical workflows. This review aims to delve into the theoretical underpinnings of radiogenomics and explore its practical applications in the diagnosis, management, and treatment of BC and to put radiogenomics on a path towards fully integrated diagnostics.

## 1. Introduction

Cancer affects people of all ages and races worldwide and results from genetic changes caused by environmental and genetic factors [1]. Breast cancer (BC) is among the most common causes of morbidity and mortality among women globally. Early detection and precise treatment of BC is crucial, with treatment of advanced stage cancer showing poorer survival rates. Genomic and environmental differences between patients with European versus African ancestry have been shown in many studies [2,3,4,5]. Patients of African ancestry have lower survival rates and poorer outcomes for BC globally [6]. Factors influencing this include ethnicity, biology, as well as sociocultural and socioeconomic factors [6]. Pinheiro et al. analyzed cancer mortality data collected from south Florida between 2012 and 2016 amongst Hispanic, White, and Black populations with further sub-categorizations for African-American, Cuban, Puerto Rican, Afro-Caribbean, and South American groups [7]. The study showed that mortality rates were the highest for the African-American sub-population [7]. Cancer studies in a local setting are pivotal to provide optimal diagnosis, prognosis, and treatment in SA.

Breast cancer is the most common cancer diagnosed in women worldwide, accounting for 2.3 million new diagnoses and 685,000 deaths in 2020 [8]. Despite recent improvements in its detection, treatment, and prevention, BC is still a difficult disease to manage, impacting millions of people globally. Additional advances in clinical care and scientific knowledge are required to improve the quality of life and survival rates for individuals with this condition.

Pre- and post-menopausal women have been shown to have different burdens of BC subtypes, with younger women having higher rates of triple-negative BC and older women a higher proportion of ER-positive luminal BC [9,10]. BC diagnosis, prognosis, and treatment have changed as a result of improved knowledge of the molecular heterogeneity inherent in many subtypes of BC [11]. Histological subtypes of BC include ductal, lobular, papillary, mucinous, tubular, and medullary cancers [11,12]. Targeted therapy has improved with further sub-classification of BC according to molecular characterization. Luminal A, Luminal B, basal-like (triple negative), HER2-enriched, normal breast-like, and Claudin-low are among the molecular subtypes of invasive BC [13]. This classification is based on the expression of the estrogen receptor (ER), progesterone receptor (PR), and human epidermal growth factor receptor 2 (HER2), as well as gene expression profiling.

Numerous studies have shown that treatment outcomes are poorer and survival rates are lower when BC is diagnosed at the later stages of cancer growth [14,15,16,17]. Thus, it is imperative that BC is diagnosed early and characterized precisely. In some resource-restricted settings, there is a lack of adequate histopathology expertise, and in such scenarios, emerging technology can help fill this gap. This includes radiogenomics, a new and rapidly expanding application of precision medicine that combines data from genomics and radiomics.

This review will examine the use of radiogenomics in the diagnosis and management of BC.

## 2. Triple Approach to Breast Imaging and Medical Imaging Techniques

The triple approach to breast imaging is used in practice for the detection of breast masses. The triple approach includes clinical examination, imaging such as breast ultrasound or mammography, and biopsy [18,19]. The common imaging techniques used in breast cancer diagnostics and observations include magnetic resonance imaging (MRI), computed tomography (CT), and positron emission tomography/computed tomography (PET/CT).

Magnetic resonance imaging is a non-invasive procedure that can produce precise images of the body using non-ionizing electromagnetic radiation and radio frequency (RF) radiation in the presence of controlled magnetic fields [20]. The results can be used to make diagnoses, design therapeutic strategies, and evaluate treatment response and disease progression.

Computed tomography is a widely used technology for cross-sectional imaging, which uses ionizing radiation or x-rays along with an electronic detector array to capture the patterns of densities [21].

Positron emission tomography (PET) evaluates tissue and organ function by using minuscule amounts of radioactive chemicals or radiopharmaceuticals injected into the bloodstream of a patient. PET/CT scans can be used to diagnose BC in its early stages by spotting cellular alterations. 18Fluorine-fluorodeoxyglucose (FDG) PET-CT uses fluorine-18 labelled FDG, a glucose analog, to stage BC and monitor treatment response. Most malignancies have rapid glycolysis, which is exploited by FDG-PET imaging, as a result of the over-expression of glucose transporters (GLUTs) and increased hexokinase activity [22,23].

## 3. Genomics

Giving the patient the appropriate treatment in the proper dosage at the appropriate time based on both their germline genetic makeup and the somatic genetic makeup of their tumor is the aim of precision medicine when applied to cancer [24]. Open-access databases that enable researchers to access and participate in the curation of vast amounts of genomic data include the NCI Genetic Data Commons [24], GWAS catalog [25], ClinGen [26], and ClinVar [27].

### Sequencing

Sequencing can be used to identify how the molecular building blocks are arranged within the tissue of interest. As a result of the advancement of next-generation sequencing (NGS) technologies, it is possible to sequence the entire human genome in less than a day for less than USD 1000 per genome [28,29], and it is the commonly used sequencing method. The advantages of NGS, a parallel sequencing method, include speed, exceptionally high throughput, and scalability [28].

## 4. Radiogenomics and Its Use in Precision Medicine

Precision medicine is a new method for treating and preventing diseases that considers a person’s unique genetics, environment, and lifestyle. Numerous studies are looking for a technique to improve cancer diagnosis, prognosis, treatment efficacy, and survival rates. Researchers and clinicians are searching for non-invasive imaging biomarkers that can be connected to clinical outcomes and genomic characteristics [11]. Radiogenomics integrates genomic profiles and medical imaging features identified above and has shown promise in the diagnosis and prognosis of cancer [30,31]. It has the ability to combine sizable amounts of quantitative data from medical images with specific genomic characteristics and build predictive models. As such, radiogenomics seeks to provide a deeper understanding of tumor biology [1,32]. The ultimate aim is to enhance cancer outcomes by creating imaging biomarkers that include phenotypic and genotypic elements. The non-invasive nature of medical imaging, relative cost-effectiveness, and potential for early cancer detection make radiogenomics popular [33]. These features and images can then be utilized to infer the presence of genetic biomarkers and make detailed decisions on prognosis and therapy. Understanding each component separately and how it might be integrated is crucial, because radiogenomics seeks to combine the predictive power of radiomics and genomics where the features within the medical image can be used to infer the genomic features of the patient. Radiomics is described as “high throughput quantitative feature extraction that results in the conversion of images into mineable data” [31,34]. The association of specific features with the presence of specific genomic patterns means that these data can be used to assess a person’s health, a condition, or treatment response [32,33]. The very basic radiogenomic study workflow used to identify the association between radiomic and genomic features is shown in Figure 1.

### 4.1. Acquisition of Raw Images

The images are acquired through CT, MRI, or PET/CT scans. The structural and functional characteristics of a tumor are revealed by PET/CT and single photon emission CT (SPECT). The molecular imaging method FDG-PET identifies changes in the metabolic activity within the tumor. The rate of absorption, metabolism, and accumulation can be utilized to evaluate the effectiveness of treatments and the course of the disease [35,36,37,38].

### 4.2. Pre-Processing of Information

It is crucial to pre-process the raw image data in order to support the uniform and trustworthy identification of characteristics. The Region of Interest’s (ROI) imaging signals can be filtered. In terms of image segmentation, the consensus typically acknowledges that image segmentation divides an image into smaller regions or objects based on its elements and this is dependent on the tissue of interest [39,40]. The precision of the image segmentation phase will heavily influence the efficacy and efficiency of the subsequent stages in image processing [41]. Ren et al. demonstrated a promising Gaussian kernel probability-driven slime mould algorithm with new movement mechanism for multi-level image segmentation [41]. Ru et al. also designed an attention-guided neural ODE network for breast tumor segmentation that allowed deep neural network challenges such as overfitting, increased amounts of parameters, and lack of interpretability to be overcome [42]. Both have shown good performance in segmentation.

Manual segmentation, however, is frequently employed. If the ROI is very limited, it is unable to offer sufficient information. On the other hand, a large ROI may produce bias due to the tumor’s heterogeneity throughout its volume. Full manual segmentation is preferred by some doctors, although it takes a long time and may reveal inter-observer variability [43,44]. The effectiveness of automatic segmentation depends on the algorithm’s precision and capacity to distinguish the ROIs from the surrounding tissues. Automatic segregation can be managed by a variety of technologies [45,46,47]. Because of this, numerous studies have demonstrated that semi-automatic segmentation is a more suitable method [48].

Um et al. extracted 420 characteristics from 161 cases using five image pre-processing approaches [49]. The diversity of radiomic features was most significantly reduced by histogram standardization. The findings demonstrated that patients can be categorized according to their chances of survival [49]. In order to provide repeatable and precise segmentations, Veeraraghavan et al. created a unique semi-automatic method by merging the multiparametric Gaussian Mixture Model (GMM) and GrowCut (GC) for cancer [50]. Approximately 75 patients with invasive BC were used as a sample, and segmentation results were compared. When compared to hand delineations and other examined segmentation approaches, GrowCut Gaussian Mixture Model (GCGMM) segmentations have been demonstrated to be more repeatable [50].

### 4.3. Extraction of Features

Feature selection typically involves employing a candidate algorithm to identify the best feature sets during pre-processing in machine learning and data mining. This process optimizes the dataset’s features for analysis and enhances classification performance using the selected optimal feature combination [51]. The extraction of high-dimensional feature sets is crucial for radiomics. These sets can be used to create prediction models and to define the characteristics of cancer phenotypes. Specialized software can process radiomic characteristics, which includes PyRadiomics [52,53], Comprehensive Evaluation of Relapse Risk (CERR) [54,55], or Imaging Biomarker Explorer (IBEX) [56,57]. The volume, surface area, and sphericity of three-dimensional (3D) objects can be collected using morphology-based attributes. The ROI’s grey-level distribution can be assessed using intensity-based features, which can also describe the overall intensity variability (first order) and the local distribution (second order). These are referred to as “Texture features”. The Grey Wolf Optimization (GWO) technique is a commonly employed meta-heuristic approach; however, its capability for searching is restricted when addressing most function optimization problems. For this reason, a new variant of GWO, called SCGWO, was proposed by Hu et al., combining GWO with an enhanced spread strategy and a chaotic local search (CLS) mechanism to overcome these performance limitations [58]. Specifically, a spread strategy was introduced into the basic GWO to modify the search agent’s ability to enhance global exploration capability, increase randomness in individual movements, and evade local optima. Subsequently, a CLS mechanism was adopted to expedite the convergence rate of the evolving agents [58].

A hybridization model for selecting an optimal feature subset was also developed by Xu et al. The model is achieved through an innovative binary version of the k-nearest neighbor classifier (KNN) and the moth-flame optimizer for classification tasks. The technique, referred to as MFeature or ESAMFO, implemented various strategies, including two types of transfer functions, an ensemble strategy, a simulated annealing (SA) disturbance mechanism, and a crossover scheme. These strategies aimed to enhance the equilibrium between global exploration and local exploitation capabilities within the basic MFO framework [59].

For the examination of intra-tumoral heterogeneity in pathology, sophisticated texture analysis is being developed. The relationship between the placement within ROIs and the grey-level intensity of pixels can be examined using texture analysis. Extraction, texture discrimination, texture classification, and form reconstruction are the four phases that comprise texture analysis [60,61]. Dynamic contrast-enhanced CT or MRI, as well as metabolic PET, are utilized to obtain dynamic characteristics that are used to assess tracer uptake in tumors over time. This can reveal details about the connections between tumor prognosis and molecular subclassifications [62].

### 4.4. Data Analysis

The variables and features gathered during extraction may be redundant or contain extraneous data. To go through all the data and retain only the important data, data selection or analysis is required. The filter, wrapper, and embedding approaches are the three most used selection models. Without using a model, the filter methods can evaluate features. Predictor optimization is a step in the selection process for wrapper models [63]. The wrapper models deliver better outcomes. The filtering techniques, however, are less expensive. The learning component and feature selection are combined in the embedded techniques [63]. In the filter method, the feature selection process is unrelated to the choice of classifier, as it depends on the overall characteristics of the training data for feature selection, irrespective of any specific predictor. The wrapper models encompass optimizing a predictor within the selection process, often yielding superior results, but it is worth noting that filter methods are typically more computationally efficient compared to wrappers. The embedded methods inherently integrate the learning and feature selection components, making it impossible to disentangle them [63].

Deep learning uses convolutional neural networks (CNNs). Through a set of layers, CNN blends image filters with artificial neural networks [64]. CNNs use local connections and weights to assess the input images. The pooling operations to acquire spatially invariant features are then performed [65]. Algorithms can determine the ideal feature set and the significance of each feature after obtaining enough training data [66]. A prediction model is needed to link the chosen features with the genetic data once the set has been received. The likelihood of clinical use is confirmed using a radiomic model. An investigation into radiogenomics may be hypothesis- or exploratory-driven. A multiple hypothesis test is frequently used in exploratory investigations, where features retrieved are compared to other genetic variables. Using the hypothesis-driven method, researchers gather enough imaging phenotypes and conduct their research with a particular hypothesis in mind [67].

Medical imaging analysis is now being developed using Artificial Intelligence (AI) and initial results are promising. In terms of cancer diagnosis and prognosis, it is thought that AI can possibly match or even enhance work conducted by skilled pathologists [68]. AI-powered radiogenomics will be able to identify patterns in an image and associate those traits with particular phenotypes [69]. Changes at the genetic, transcriptomic, translational, and epigenomic levels can all be linked to the phenotypic features present in medical images. The prognostic, diagnostic, and therapeutic techniques can all be improved using this knowledge [70]. These image characteristics can also be employed as predictors and markers of survival [69]. Radiogenomics and AI work together to increase knowledge and accuracy for better diagnosis, prognosis, and treatment.

## 5. Current Application of Radiogenomics in Oncology

Using large data analysis techniques, radiogenomics offers a comprehensive understanding of tumor biology and imaging biomarkers [71]. These methods have been proven effective in a number of tumor types [31]. There is evidence to support a relationship between cancer genetic features and imaging [72,73,74,75,76,77,78,79,80,81,82,83,84,85,86,87,88,89,90]. MRI predominates in radiogenomics for breast imaging [91] and has been found to be the most accurate test for finding BC [92,93,94]. Yamamoto et al. looked at 10 patients who had preoperative dynamic contrast-enhanced (DCE)-MRI and global gene expression data [95,96,97,98,99,100,101,102,103,104,105,106,107,108,109,110,111,112,113,114,115,116,117,118]. The relationship between MRI phenotypes and underlying global BC gene expression patterns was presented using a preliminary radiogenomic association map. In patients with BC, high-level analysis found 21 imaging features associated with 71 percent of the total number of genes tested. Significant relationships existed between the diverse enhancement patterns and the interferon BC subtype [95]. The same researchers also looked into the multiscale interactions between early metastasis, long non-coding RNA (lncRNA) expression, and quantitative computer vision-extracted DCE-MRI phenotypes utilizing RNA sequencing [88].

The practice of multi-modal analysis is commonly used to transfer knowledge across different modes of information. This approach, known as co-learning, links a modality with abundant resources to one with limited resources, enhancing inference capabilities in the latter through the relevant modality [96]. Multi-modal analysis has found application across diverse domains including geographical and biomedical image analysis [97,98], video analysis [99,100], and sentiment analysis [101]. Various methods facilitate co-learning in multi-modal analysis, such as tensor learning [102], generative models [103], graphical models [104,105], prior knowledge regularization [106], multiple kernel learning [107], and neural networks [108,109,110].

Recently, data representation techniques have been introduced in bioinformatics for multi-modal analysis [111,112], particularly employing the subtensor learning framework for cross-modal module discovery. This framework treats data from different modalities as tensor slices, allowing tensor decomposition to extract subtensors as latent knowledge shared among modalities [113,114].

Images with contrast medium augmentation were divided into “rim” and “entire” patterns. The rim pattern indicated that the periphery of a tumor lesion was more amplified than the center [115,116] and has led to the enhanced rim fraction (ERF) scoring system. The ERF score is linked to early metastasis and poor metastasis-free survival [33,117]. Using radiogenomics, eight lncRNAs that contributed to the EFR score were discovered. Homeobox transcript antisense intergenic RNA expression is linked to this improving rim fraction score. Early metastatic disease and low metastasis-free survival are known to be predicted by homeobox transcript antisense intergenic RNA [93]. Breast cancer risk is greater in women who have the BRCA-1 or BRCA-2 gene mutation [118]. Li et al. discovered that low-risk women can be distinguished from BRCA1/2 gene-mutation carriers using computerized mammographic assessment of parenchymal patterns and breast density using radiographic texture analysis [77].

There are some obstacles that radiogenomics in clinical practice must overcome. The repeatability and reproducibility of the present radiogenomic models is one of these issues [118]. Researchers need to consider the diversity brought on by using various hardware, software, or clinics, according to Shui et al. [1]. To guarantee the quality and dependability of analytical data in radiogenomic research, standard practice guidelines are essential [119]. Radiogenomics, however, can accelerate the development of precision oncology and individualized patient care in the future.

## 6. Limitations

Radiogenomics faces significant challenges [120]. Some studies have linked genomic data with imaging features, but not all are relevant to prognostic outcomes [113]. The complexity of gene expression and signaling pathways introduces bias when directly linking imaging features with genomic data. Moreover, radiogenomic studies are susceptible to statistical overfitting due to the difficulty of matching large genomic datasets with imaging data. Grouping individual genetic mutations into gene traits may compromise the imaging’s predictive ability. Additionally, inter-observer variation in qualitative imaging features necessitates periodic evaluation. Overall, these issues underscore the need for careful consideration and refinement in radiogenomic research [113]. Regarding the radiomic part of radiogenomics, there are differences in the acquisition parameters during image acquisition and reconstruction. The contrast enhancement protocols also vary across machines and patients. This will lead to variations in the results. Standardized protocols adapted to each modality, comprehensive parameter descriptions, and standardization of control ROIs are needed to overcome this challenge [121]. One key limitation of radiogenomics is the preparation of the radiogenomic models where image features are linked to genomic features. Molecular characterization necessitates invasive tissue collection and makes use of genomic, proteomic, and transcriptomic technologies. The biopsy sample might not be a complete representation of the lesion, being extracted from a small area of the tumor’s diverse lesion leading to false result interpretation. Due to the high expense, large-scale genomic profiling is not practical. Additionally, sorting, storing, and analysis of the data would take time, delaying the outcome. Databases and software are also restricted that can allow for missed information. To address this, semi-automatic or fully automatic segmentation and data analyses methods can be used by utilizing public resources and programs like The Cancer Imaging Archive (TCIA) and The Cancer Genome Atlas (TCGA) [113,121]. However, relying on retrospective studies has its limitations, and reproducibility is limited. Therefore, a well-designed multicenter prospective study with a large dataset should be conducted in the future to overcome these challenges and further advance research in this field [113]. Publications should include access to raw data and methods used during feature extraction and segmented ROI [121]. A repository can also be designed that can be accessible publicly. In radiogenomic studies, genomic data are primarily sourced from microarray data, with limited focus on microRNAs. Given their capacity to target numerous genes and regulate gene expression, exploring correlations between microRNAs and imaging data represents a promising new direction in research. The immunohistochemistry images needed for correlation and comparison with other data are not present in these databases [122]. The ideal database should contain genomic data (mutations, copy number variations, etc.), radiomic feature data, epigenomic (non-coding RNAs), metabolomic data, proteomic data, immunohistochemical data, and transcriptomic data [123]. Artificial intelligence can assist in the handling and mining of data. Additionally, the image needs to be uniform. The definition of reproducible and stable radiogenomic biomarkers also requires the use of biochemical techniques.

## 7. Conclusions

Emerging research has shown that radiogenomics has made tremendous progress in precision oncology and is favored since it is non-invasive [1,124,125,126]. The use of radiogenomics and its emphasis on the connection between genomics and imaging phenotypes have been demonstrated [120,124,127]. Radiogenomic analysis is particularly useful in cases where there is a lack of data [120]. By identifying genomic information and imaging that have a high potential for outcome prediction, radiogenomics can direct the collection of future datasets [117]. Despite its limitations, radiogenomics has the potential to contribute to the development of precision oncology and individualized patient treatment. Radiogenomics has the potential to eliminate the need for biopsy and sequencing. Breast cancer radiogenomics is a promising area of research that can benefit from advances in genetics and data processing [1,128]. The majority of radiogenomic research is currently concentrated on retrospective datasets and solitary institutions [128]. Prospective research and results’ validation are required in conjunction with multi-institutional settings in order to enhance patient care and realize the full potential of radiogenomics [117]. For methods and analyses to be standardized and optimized, more study is required. AI can potentially be utilized to improve the results’ accuracy and enable the use of radiogenomics in clinical settings [123,129].

## Figures and Tables

**Figure 1 cancers-16-01076-f001:**
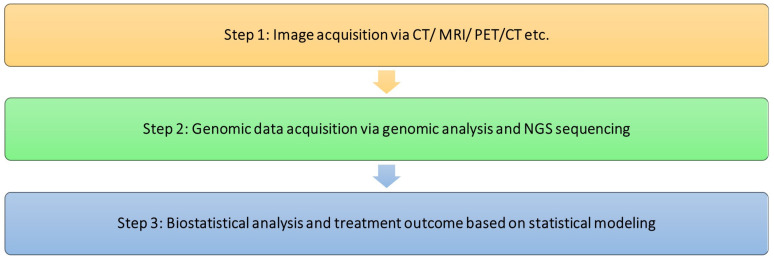
The basic radiogenomic study workflow used to identify the association between radi-omic and genomic features.

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
