# Peer review of "The Convergence of Radiology and Genomics: Advancing Breast Cancer Diagnosis with Radiogenomics"

_cancers, 2024, doi:10.3390/cancers16051076_

Round 1

Reviewer 1 Report

Comments and Suggestions for Authors

Dear Authors,

The manuscript is well written, has important clinical message, and should be of great interest to the readers.

I have just a few suggestions.

I think that the breast imaging paragraphs are full of details, useless to the comprehension of the main messagge, namely radiogenomics (and not breast imaging diagnostic techniques). I suggest to shorter it or even delete it.

Aditionally, I
suggest to include a picture/diagram to illustrate the main step of radiogenomic worflow study (instead of figure 2) to make the paper more appealing for readers.

Author Response

Reviewer 1:

Comment 1

I think that the breast imaging paragraphs are full of details, useless to the comprehension of the main messagge, namely radiogenomics (and not breast imaging diagnostic techniques). I suggest to shorter it or even delete it.

Response

2.1 Breast ultrasound (page 2) has been deleted.

2.2 Mammography (page 3) has been deleted.

The triple approach and medical imaging were combined into section 2 (page 2). Each imaging technique was shortened.

Comment 2

Aditionally, I suggest to include a picture/diagram to illustrate the main step of radiogenomic worflow study (instead of figure 2) to make the paper more appealing for readers

Response

Figure 2 has been replaced with a very basic image showing the steps of radiogenomics. Extra information was removed to decrease lengthy discussion

Reviewer 2 Report

Comments and Suggestions for Authors

This manuscript aims to explore its application in the diagnosis and management of breast cancer through an introduction to the theoretical basis of radiogenomics, and analyze its limited and future development directions. About half of the manuscript is used to introduce the theoretical basis of the relevant topics, the application of radiogenomics in the diagnosis and treatment of breast cancer is introduced in Section 6, and the relevant limitations are analyzed in Section 7. Overall, the manuscript needs to be significantly changed to meet the needs of publication.

1. It is recommended to reduce the introduction of basic theories. Although this section is necessary for radiogenomics, it is not recommended to devote a lot of space to it. This part of the content, as a basic theory, is also suitable for the application of radiogenomics in other diseases.

2. Add the content of Section 6, in the diagnosis and treatment of breast cancer, radiogenomics have a large number of applications in different problems, such as disease diagnosis, prognosis prediction, management, etc. It is recommended to increase and refine the application of radiogenomics in these areas.

3. For the content of Section 7, it is recommended to correspond to the revised Section 6 and introduce in more detail the limitations of the radiogenomics on different issues.

4.Section 5 of the text, which has only subsection 5.1 and no subsection 5.2, it is suggests that the numbering of the subsection in the text be adjusted.

Comments on the Quality of English Language

The structure of the manuscript is suggested to be adjusted in accordance with the previous comments, and the language is clearly expressed

Author Response

Reviewer 2:

Comment 1.

 It is recommended to reduce the introduction of basic theories. Although this section is necessary for radiogenomics, it is not recommended to devote a lot of space to it. This part of the content, as a basic theory, is also suitable for the application of radiogenomics in other diseases.

Response

Lengthy information was cut to be more precise to radiogenomics. Radiomic and genomic aspects are discussed very briefly.

Comment 2

Add the content of Section 6, in the diagnosis and treatment of breast cancer, radiogenomics have a large number of applications in different problems, such as disease diagnosis, prognosis prediction, management, etc. It is recommended to increase and refine the application of radiogenomics in these areas

Response

Section 6 was revised and is now section 5. Multi-model and cross-modal were discussed briefly. The section was revised to remove lengthy discussion.

Comment 3

For the content of Section 7, it is recommended to correspond to the revised Section 6 and introduce in more detail the limitations of the radiogenomics on different issues.

Response

Limitations has been revised. Possible solutions were also included. The impact is indicated in blue and the solution in green.

Comment 4

Section 5 of the text, which has only subsection 5.1 and no subsection 5.2, it is suggests that the numbering of the subsection in the text be adjusted.

Response

This section was removed during the revision process

Reviewer 3 Report

Comments and Suggestions for Authors

Suggestions: Minor Revisions

Score:80

This article explores significant issues in the field of breast cancer (BC) and introduces radiogenomics as a promising area in precision medicine. By associating genetic patterns with image attributes, radiogenomics is poised to significantly enhance early detection, prognosis, and diagnosis of BC. Overall, this review aims to delve into the theoretical foundations of radiogenomics and explore its practical applications in the diagnosis, management, and treatment of breast cancer.I have a few comments that need addressing by the authors:

1.Suggest optimizing the article structure, especially in the sections introducing medical imaging and genomics workflow, to ensure a tighter relevance of content. Consider reducing lengthy descriptions in these sections and focus more on the aspects directly related to breast cancer.

2.When discussing the current application status of radiogenomics in oncology, it is recommended to introduce research on cross-modal fusion for tumor analysis. Multi-modal fusion utilizes complementary information between different modalities, aiming to achieve better precision in the analysis of breast cancer for accurate medical outcomes.

3. It is recommended to use the algorithm in ‘Gaussian kernel probability-driven slime mould algorithm with new movement mechanism for multi-level image segmentationand Attention guided neural ODE network for breast tumor segmentation in medical images for introduction of the image segmentation stage in the part of Pre-Processing of Information, as it also has good performance in segmentation.

4. Consider adding ‘Chaotic diffusion-limited aggregation enhanced grey wolf optimizer: Insights, analysis, binarization, and feature selection.’ and ‘MFeature: Towards High Performance Evolutionary Tools for Feature Selection’ in introduction of the feature extraction and feature selection to enrich the feature selection research background.

5. In discussing the limitations section, it is recommended that the authors provide a clearer discussion in the article about the potential impact of this limitation on the research results and consider exploring possible solutions. Are there existing technologies or methods that can address the incompleteness of the data? Authors are encouraged to consider the limiting factors in experimental design and provide some practical suggestions.

Author Response

Reviewer 3:

Comment 1.

Suggest optimizing the article structure, especially in the sections introducing medical imaging and genomics workflow, to ensure a tighter relevance of content. Consider reducing lengthy descriptions in these sections and focus more on the aspects directly related to breast cancer.

Response

Lengthy information was cut to be more precise to radiogenomics. Radiomic and genomic aspects are discussed very briefly.

Comment 2

When discussing the current application status of radiogenomics in oncology, it is recommended to introduce research on cross-modal fusion for tumor analysis. Multi-modal fusion utilizes complementary information between different modalities, aiming to achieve better precision in the analysis of breast cancer for accurate medical outcomes.

Response

Multi-model and cross-modal were discussed in newly revised section 5 and is highlighted in yellow. This is briefly noted.

Comment 3

. It is recommended to use the algorithm in ‘Gaussian kernel probability-driven slime mould algorithm with new movement mechanism for multi-level image segmentation’ and ‘Attention guided neural ODE network for breast tumor segmentation in medical images’ for introduction of the image segmentation stage in the part of Pre-Processing of Information, as it also has good performance in segmentation.

Response

This section has been cut due to the comment to shorten lengthy descriptions.

Comment 4

Consider adding ‘Chaotic diffusion-limited aggregation enhanced grey wolf optimizer: Insights, analysis, binarization, and feature selection.’ and ‘MFeature: Towards High Performance Evolutionary Tools for Feature Selection’ in introduction of the feature extraction and feature selection to enrich the feature selection research background.

Response

This section has been cut due to the comment to shorten lengthy descriptions.

Comment 5

 In discussing the limitations section, it is recommended that the authors provide a clearer discussion in the article about the potential impact of this limitation on the research results and consider exploring possible solutions. Are there existing technologies or methods that can address the incompleteness of the data? Authors are encouraged to consider the limiting factors in experimental design and provide some practical suggestions.

Response

Impact of limitations had been highlighted in blue and possible solutions have been highlighted in green. More information has been added to the limitation section.

Round 2

Reviewer 2 Report

Comments and Suggestions for Authors

There are still some formatting problems in the manuscript that need to be revised, and after the revision, the manuscript can be considered for acceptance.

Comments on the Quality of English Language

The structure of the paper is relatively clear, the expression is accurate, and the sentences are clear.